# Enantioselective Binding of Proton Pump Inhibitors to Alpha1-Acid Glycoprotein and Human Serum Albumin—A Chromatographic, Spectroscopic, and In Silico Study

**DOI:** 10.3390/ijms251910575

**Published:** 2024-10-01

**Authors:** Gergely Dombi, Levente Tyukodi, Máté Dobó, Gergely Molnár, Zsuzsanna Rozmer, Zoltán-István Szabó, Béla Fiser, Gergő Tóth

**Affiliations:** 1Department of Pharmaceutical Chemistry, Semmelweis University, Hogyes E. 9, 1092 Budapest, Hungary; dombi.gergely@stud.semmelweis.hu (G.D.); dobo.mate@stud.semmelweis.hu (M.D.); molnar.gergely@stud.semmelweis.hu (G.M.); 2Center for Pharmacology and Drug Research & Development, Semmelweis University, 1085 Budapest, Hungary; 3Department of Pharmaceutical Chemistry, University of Pécs, 7624 Pécs, Hungary; tyukodi.levente@pte.hu (L.T.); rozmer.zsuzsanna@pte.hu (Z.R.); 4Department of Pharmaceutical Industry and Management, Faculty of Pharmacy, George Emil Palade University of Medicine, Pharmacy, Science, and Technology of Targu Mures, Gh. Marinescu 38, 540142 Targu Mures, Romania; zoltan.szabo@umfst.ro; 5Sz-Imfidum Ltd., 525401 Lunga nr 504, 525401 Targu Mures, Romania; 6Institute of Chemistry, University of Miskolc, 3515 Miskolc, Hungary; bela.fiser@uni-miskolc.hu; 7Department of Biology and Chemistry, Ferenc Rakoczi II Transcarpathian Hungarian College of Higher Education, Transcarpathia, 90200 Beregszasz, Ukraine; 8Department of Physical Chemistry, Faculty of Chemistry, University of Lodz, 90-149 Łódź, Poland

**Keywords:** AGP, HSA, proton pump inhibitors, protein binding, HPLC, transport proteins

## Abstract

The enantioselective binding of three proton pump inhibitors (PPIs)—omeprazole, rabeprazole, and lansoprazole—to two key plasma proteins, α1-acid glycoprotein (AGP) and human serum albumin (HSA), was characterized. The interactions between PPI enantiomers and proteins were investigated using a multifaceted analytical approach, including high-performance liquid chromatography (HPLC), fluorescence and UV spectroscopy, as well as in silico molecular docking. HPLC analysis demonstrated that all three PPIs exhibited enantioseparation on an AGP-based chiral stationary phase, suggesting stereoselective binding to AGP, while only lansoprazole showed enantioselective binding on the HSA-based column. Quantitatively, the *S*-enantiomers of omeprazole and rabeprazole showed higher binding affinity to AGP, while the *R*-enantiomer of lansoprazole displayed greater affinity for AGP, with a reversal in the elution order observed between the two protein-based columns. Protein binding percentages, calculated via HPLC, were greater than 88% for each enantiomer across both transport proteins, with all enantiomers displaying higher affinity for AGP compared to HSA. Thermodynamic analysis indicated that on the HSA, the more common, enthalpy-controlled enantioseparation was found, while in contrast, on the AGP, entropy-controlled enantioseparation was observed. The study also identified limitations in using fluorescence titration due to the high native fluorescence of the compounds, whereas UV titration was effective for both proteins. The determined log*K* values were in the range of 4.47–4.83 for AGP and 4.02–4.66 for HSA. Molecular docking supported the experimental findings by revealing the atomic interactions driving the binding process, with the predicted enantiomer elution orders aligning with experimental data. The comprehensive use of these analytical methods provides detailed insights into the enantioselective binding properties of PPIs, contributing to the understanding of their pharmacokinetic differences and aiding in the development of more effective therapeutic strategies.

## 1. Introduction

Transport proteins are integral components of biological systems, facilitating the movement of molecules across cellular membranes and within extracellular spaces. Among these proteins, human serum albumin (HSA) and α1-acid glycoprotein (AGP) stand out as key players in the transport and distribution of a diverse range of endogenous and exogenous compounds within the bloodstream [1,2]. HSA, the most abundant protein in human plasma, comprises approximately 60% of the total plasma protein content. Structurally, HSA is a single-chain polypeptide consisting of 585 amino acids organized into 3 homologous domains. According to Sudlow’s classification, HSA features two primary binding sites for drug ligands: Sudlow site I, situated within subdomain IIA, and site II, located in subdomain IIIA. Site I predominantly facilitates the selective binding of heterocyclic anions, such as ibuprofen, while aromatic carboxylates bind to site II. However, it is important to recognize that not all ligand binding conforms to Sudlow’s model. Site III, an additional binding pocket found in subdomain IB, forms a hydrophobic D-shaped cavity, and is utilized for interaction by certain compounds, such as bilirubin. Several studies have proposed Sudlow site III as a third major binding site with the potential to accommodate drug ligands on HSA [3,4]. Nevertheless, it is worth noting that differences may exist between in vitro and in vivo binding data, considering factors such as the plasma’s fatty acid composition, significantly influencing binding properties [5]. The unique structure endows HSA with exceptional versatility and binding ability, allowing it to interact with various ligands, including fatty acids, hormones, metal ions, and drugs. Through reversible and non-covalent interactions, HSA acts as a carrier for numerous hydrophobic and hydrophilic molecules, influencing their pharmacokinetic profiles and tissue distribution [6,7,8,9,10]. In contrast, AGP, also known as orosomucoid, is a glycoprotein synthesized primarily by the liver and secreted into the bloodstream. Although present in lower concentrations compared to HSA, AGP plays a significant role in drug binding and transport, particularly during acute-phase responses and inflammatory conditions. In healthy individuals, the basal plasma concentration of AGP is typically around 20 μM. However, in conditions characterized by stress, such as sepsis, this concentration can surge dramatically, increasing up to five-fold [11]. Consequently, the impact of AGP binding on the pharmacological activity of highly bound drugs can be substantial during acute-phase reactions, such as sepsis. Structurally, AGP is characterized by a single polypeptide chain with extensive glycosylation. The protein’s acidity, with an isoelectric point of 2.7, primarily stems from the presence of sialic acid residues, imparting a net positive charge to the protein. This unique charge distribution allows AGP to preferentially bind basic and neutral drugs, including many lipophilic substances and certain pharmaceutical agents. Genetic polymorphism of AGP has long been recognized, with three major variants (F1, S, and A), distinguished by differences in their primary structure. The distinction between F1 and S arises from a single amino acid substitution, while the A variant differs from F1/S by 22 residues. Commercially, pooled-source protein typically consists of approximately 40% F1, 30% S, and 30% A variants. Notably, there exists a significant disparity in the binding of certain drugs to the F1/S and A variants of AGP [11,12,13,14].

There are numerous methods available for characterizing protein–drug interactions, each offering unique insights into the nature of these interactions, as each has different advantages and disadvantages [15]. Spectroscopic methods, such as circular dichroism, fluorescence, UV-Vis, and nuclear magnetic resonance spectroscopy, are commonly employed methods for studying protein–drug interactions [16]. Fluorescence spectroscopy is a sensitive method for measuring the complex stability between analytes and ligands. Changes in the fluorescence intensity or emission wavelength indicate the changes in the three-dimensional structure of proteins [17]. Spectroscopic approaches are most effective for high-affinity binding sites and less useful for studying multiple equilibria. This is because the analytical response does not directly measure the extent of binding, but rather is proportional to it. The other problem for the determination of the binding constant could be if the ligand has its own fluorescence, and thus, it can have an impact on the exact determination. This can be overcome by considering the inner filter effect [18,19]. Chromatographic or electrophoretic methods are also widely used for the characterization of protein–analyte binding [20,21,22,23]. High-performance affinity chromatography is an often-used technique that involves the immobilization of a protein onto a support matrix and the subsequent injection of an interacting solute into the column. In this process, drugs exhibiting significant affinity will interact with the immobilized protein. Higher interaction comes together with high retention on the column, while compounds with minimal binding will elute earlier. Based on this, the binding percentage could be calculated. The immobilization of the protein on the chromatographic support is the main debated and critical aspect to consider when performing binding studies with affinity chromatography. The immobilization process can alter the three-dimensional structure of the protein and can lead to incorrect results [15,24]. Additionally, ultrafiltration [25], ultracentrifugation [26], calorimetric techniques [27], and UV-pH titration [28] are also utilized to elucidate the binding kinetics and thermodynamics of these interactions. In recent years, computational methods have emerged as valuable tools for characterizing stereoselective drug–protein interactions. Understanding analyte–protein binding interactions at the atomic level can also inform drug design efforts, easing the development of more effective and selective therapeutic agents [29,30]. Characterizing protein–ligand interactions can only be effectively accomplished using a multi-analytical approach, which takes the advantages of each analytical method and grants that the results obtained are validated through different techniques. For the rapid characterization of enantioselective binding, methods such as HPLC or electrophoresis can be applied. Spectroscopic techniques are suitable for determining stability constants, while computational modeling is used to describe interactions at the atomic level.

The different binding characteristics of enantiomers can lead to significant variations in a drug’s absorption, distribution, metabolism, and excretion (ADME), as well as its therapeutic efficacy and safety profile. It is well established that enantiomers can exhibit distinct therapeutic effects: a classic example is thalidomide, where R-thalidomide has sedative effects, while S-thalidomide is teratogenic. In addition to enantioselective binding to the target molecule, enantioselective binding to transport proteins is also important because it provides insights into how different enantiomers are distributed, transported, and cleared in the body. This understanding enables the selection of the most effective and safe enantiomer, optimizing therapeutic outcomes and minimizing potential side effects.

Proton pump inhibitors (PPIs), such as omeprazole, lansoprazole, and rabeprazole, exert a potent and sustained reduction in stomach acid production. This effect is achieved by irreversibly inhibiting the H^+^/K^+^ ATPase proton pump located in the stomach lining. Thus, PPIs effectively suppress the secretion of gastric acid into the stomach lumen, thereby alleviating symptoms associated with conditions such as gastroesophageal reflux disease, peptic ulcers, and other acid-related disorders [31]. The unique structural characteristic of PPIs is attributed to the chiral sulfoxide functional group, which allows them to exist in two enantiomeric forms. Esomeprazole, derived from the *S*-enantiomer of omeprazole, exemplifies one of the most successful instances of a chiral switch in drug development. While dexlansoprazole (*R*-lansoprazole) and dexrabeprazole (*R*-rabeprazole), the eutomers of lansoprazole and rabeprazole, respectively, are available in some countries, their market share remains relatively modest. Interestingly, the therapeutic advantage of esomeprazole over omeprazole has been questioned in certain studies [32,33]. It is noteworthy that all PPIs are prodrugs, undergoing conversion to the non-chiral active sulfenamide or sulfenic acid form within parietal cells before binding to the enzyme. The main difference between the enantiomers is evident in their pharmacokinetic profiles, due to their enantioselective transport and metabolism [34,35,36,37]. Kim et al. investigated the enantioselective pharmacokinetics of lansoprazole and established that the effect of CYP2C19 genetic polymorphism on the enantioselective disposition of lansoprazole is not as significant as the enantioselective plasma protein binding [38].

Earlier studies have explored the binding of PPIs to BSA [39,40] and HSA [41,42,43], yet the majority of these investigated the binding of the racemic mixture, despite the well-known phenomenon of enantioselective binding [43,44]. When examining racemic compounds, we obtain an average value, which cannot be attributed to either enantiomer. The main techniques that the previous studies employed are fluorescence spectroscopy and in silico modeling. Work regarding AGP binding is limited, and only the AGP-omeprazole complex has been investigated so far [44]. Some aspects of previous studies are summarized in Table 1. Only two works deal with the enantioselective protein binding of PPIs. Karlsson and Hermansson developed a chiral HPLC method for enantioseparation of omeprazole and its metabolite on a chiral AGP-based column. Based on the successful enantioseparation, enantioselective binding can be implied; however, the study was not designed to map the binding of the studied enantiomers to the drug-binding transport protein [44]. In another work, Xu et al. developed a method for the separation of omeprazole enantiomers by affinity capillary electrophoresis using HSA as a chiral selector, clearly indicating that HSA binding of the analyte is enantioselective, which was further proven by fluorescence measurements [43]. The authors also investigated the location of the binding site through displacement studies using warfarin and ibuprofen as markers for subdomain IIA (Sudlow site I) and subdomain IIIA (Sudlow site II). Based on the results, esomeprazole is more likely to bind to Sudlow site I in subdomain IIA. However, it appears that *R*-omeprazole may bind to a different region of the protein as well. A comparison of our results with those from previous literature can be found in Appendix A.

The objective of this study was to examine the stereoselective interactions between omeprazole, lansoprazole, and rabeprazole (see the structures in Figure 1), and the primary human transport proteins, HSA and AGP. To ensure reliable results, multimodal techniques, including chromatographic, spectroscopic, and molecular modeling approaches, were employed. By elucidating the interactions between chiral sulfoxides and transport proteins at the atomic level, this research contributes to the understanding and enantioselective pharmacokinetic parameters of PPIs. Furthermore, it holds potential for informing new drug development efforts.

## 2. Results and Discussion

### 2.1. HPLC Study

HPLC using a protein-based chiral stationary phase is a fully automated method for obtaining rapid information about the binding of compounds to the respective proteins. Enantioseparation occurs in the case of unequal binding constants of the individual enantiomers to the immobilized proteins, thus allowing enantioselective binding to be characterized even when a racemic mixture was injected [9,45]. As a preliminary step, the enantiospecific binding was investigated on columns containing AGP and HSA in various alcohol:buffer mixtures. A pH 7 phosphate buffer was employed as the aqueous part of the mobile phase, while for the organic part, three different alcohols were evaluated. The findings are summarized in Table 2, while some representative chromatograms are depicted in Figure 2. On the AGP column, all enantiomers of the three compounds were successfully separated regardless of the alcohol used, while on the HSA column, only lansoprazole exhibited enantiospecific binding. It is worth mentioning that even at higher levels of organic modifier, the enantiomers of lansoprazole were well resolved on the AGP column, indicating a substantial difference in stability between the lansoprazole enantiomers with respect to AGP binding. Based on the elution order of enantiomers, it is evident that *S*-rabeprazole, *R*-lansoprazole, and *S*-omeprazole displayed a higher affinity for AGP compared to their respective antipodes. In the case of lansoprazole, a reversal of the elution order was observed between the two protein columns, since *S*-lansoprazole displayed higher affinity to HSA than *R*-lansoprazole. On the AGP column, the retention time decreased in the order of MeOH > EtOH > IPA. Although MeOH exhibited higher enantioselectivity, lower plate counts were observed, with increased retention times. Moreover, on both columns, even a small amount of organic modifier can significantly affect protein binding. Therefore, it is important to consider the type and proportion of the chosen organic modifier when comparing measured data, regardless of the applied method [46,47].

In the following step, we investigated the effect of changing some other chromatographic conditions on the protein binding. The pH of the aqueous phase influenced the protonation state of both the analyte and the proteins. It should be noted that the pH tolerance of protein-based columns is narrow due to the possible denaturation of proteins [14,48]. Taking into account the guidelines of the manufacturer, three different pH values were investigated (pH 6, 6.5, and 7) at a 95:5 (*v*/*v*%) buffer:IPA eluents ratio [49]. The buffer was uniformly 10 mM phosphate. In this range, the pH values did not have a significant effect on retention times or enantioselectivity in most cases, but the peak shape was significantly influenced. At pH 7, in general, a better peak shape was observed. One exception was lansoprazole on the AGP column, where the enantioselectivity was higher at a higher pH, but in parallel with this, the theoretical peak number was reducing. The effect of different buffer constituents was also evaluated. In addition to phosphate buffer, ammonium acetate, ammonium bicarbonate, and sodium citrate were tested and compared under the same conditions. The representative chromatograms and the principal chromatographic results are presented in Appendix A, respectively. In general, the peak shape was optimal with phosphate buffer, while the use of other salts resulted in peak distortion. Although the retention times were similar across all buffers, slight variations were observed in the resolution values. Due to the superior peak shape and adequate resolution, a phosphate buffer was selected for further studies.

#### 2.1.1. Thermodynamic Characterization of Enantioseparation and Protein Binding

The temperature effect was investigated at pH 7 and a 95:5 buffer:IPA eluents ratio, between 10 and 30 °C on both columns. Higher temperatures were not applied, as they could significantly reduce the column lifetime. The decrease in temperature led to increased retention for all molecules and increased enantioselectivity specifically for omeprazole and rabeprazole on the AGP column. Enantioseparation was only observed at low temperatures and exclusively on the AGP column for these two molecules. As a result, it was not possible to accurately calculate the thermodynamic parameters for omeprazole and rabeprazole. However, for lansoprazole, the classical van ‘t Hoff approach can be utilized to calculate thermodynamic values in both protein columns. The determination of thermodynamic parameters can provide valuable insights into the process of enantiodiscrimination, including its mechanistic aspects. The applied classical van ’t Hoff methodology postulates the existence of solely enantioselective interaction sites on the stationary phase. Nonetheless, it should be emphasized that a more realistic approach would encompass both enantioselective and non-enantioselective interactions [50,51]. The obtained thermodynamic data are summarized in Table 3. The isoenantioselective temperatures (T_iso_) were calculated by dividing Δ(ΔH°) by Δ(ΔS°). At this temperature, enthalpy and entropy compensations negate each other, resulting in the coelution of the two enantiomers and no separation. The Q values (Q = Δ(ΔH°)/T × Δ(ΔS°)_298K_) were used for visualizing the relative contribution of enthalpic and entropic terms to the free energy of adsorption. If the Q values were smaller than 1, the enantioseparation was controlled by entropy to a higher extent. If the Q values were higher than 1, it indicates that mainly the enthalpy controlled the enantioseparation, as usual.

Using an AGP column, the enantioselectivity values interestingly increased with the rising temperature. In contrast, increasing the temperature in the HSA column resulted in decreasing selectivity. The different behavior was also reflected in different thermodynamic values. It means that not only was the enantiomer elution order different, but the thermodynamics of the separation differed between the two columns. In the case of the HSA column, the usual enthalpy-controlled enantioseparation occurred, shown well by the Q value, which was higher than 1, and the high T_iso_ value. In contrast, on the AGP column, the mechanistic aspects of the enantiodiscrimination process were different. The Q value was lower than 1, and T_iso_ was also in the sub-zero range, revealing a more entropy-controlled enantioseparation. It can also be seen that in the AGP column, the ΔH° and ΔS° values were higher compared to the HSA column, but the ΔG° and Δ(ΔG°) values were similar. This example shows that thermodynamic characterization could help to explain the different enantioseparation behaviors on different protein-based columns.

#### 2.1.2. Determination of Bound Percentage between PPIs and Proteins by HPLC

Chromatographic columns containing proteins can be used to determine the percentage of drug binding [52]. A plot of the natural logarithm of the retention factors against the proportion of organic modifiers was created. This plot demonstrated a linear relationship that could be extrapolated to zero, closely simulating in vivo conditions. The intercept of these linear equations can be used to calculate the bound percentage according to Equation (4) (see in Section 3.2). The calculated bound percentage data obtained through this approach are presented in Table 4.

The protein binding percentages calculated from HPLC measurements exceeded 88% for each compound to each transport protein. The lowest binding was observed in the case of rabeprazole toward HSA. In general, the compounds had higher binding percentages for AGP than HSA. Although, the difference in the bound percentage values between the corresponding enantiomers within 1% of each other in the case of AGP enantioseparation was still achieved. Checking the effect of the organic modifier, it can also be observed that the obtained values were similar, indicating that the type of organic modifier did not significantly influence the bound percentage values. However, while the average difference between the protein binding of the enantiomers was lower than 1% for HSA, in the case of AGP, the difference was higher than 4% in some cases.

### 2.2. Spectroscopic Study

#### 2.2.1. Fluorescence Measurement

Fluorescence quenching is an excellent method for investigating binding properties because it allows rapid and reliable determination of binding affinities [53,54,55,56]. Unless there is a specific need to identify the precise binding site of a drug on a protein, quenching of the tryptophan fluorescence of proteins at 285 nm through ligand titration provides a fast and straightforward method for assessing drug binding affinities to different proteins [16]. Based on the fluorescence spectra of PPIs, we can see that their absorption peak was observed at 295 nm, which closely aligns with the typical excitation wavelength of 285 nm commonly used for protein fluorescence studies. Due to the overlapping effect in fluorescence characteristics, it has been challenging to determine the nature of the interactions between PPIs and plasma proteins using fluorescence spectroscopy alone. In principle, the fluorescence effect can be distinguished by considering the inner filter effect [18,19]. In the literature, many of the articles do not count with the inner filter effect; therefore, the results of these studies are questionable [57,58]. Some representative fluorescence spectra are depicted in Appendix A, while absorbance values used for the inner filter effect calculation and the corrected fluorescence value can be found in Appendix A. Based on our measurements, even if the inner filter effect was considered, the result was questionable, probably due to the very high fluorescence of the PPIs at 285 nm. Upon correction, it became evident that the observed emission maxima values underwent significant alterations. For instance, in the case of lansoprazole, higher values were observed at elevated concentrations compared to the AGP emission spectrum (Appendix A). This discrepancy cannot be attributed to quenching effects. In our view, due to the pronounced inner filter effect observed in this scenario, accurately calculating interactions using fluorescence quenching may not be feasible. A similar observation was published by Marciniak et al. for protopine and allocryptopine, compounds with similar fluorescence spectra as PPIs [16]. Following the instructions of this work, we determined the binding constant using the UV method.

#### 2.2.2. UV-Vis Results

The UV-Vis absorption assay is a straightforward and relatively simple method to examine the interactions between small molecules and macromolecules. Changes in the structure of the protein caused by the interacting ligand can lead to alterations in the absorption spectrum. Plasma proteins exhibited an absorption maximum at 278 nm, which was attributed to the presence of the aromatic amino acids, such as tryptophan, tyrosine, and phenylalanine. Some representative UV spectra are depicted in Appendix A. Fixed wavelength measurements were conducted at the absorption maxima of AGP and HSA. A noticeable increase in absorbance at this wavelength suggests that PPIs interacted with HSA and AGP, leading to the extension of the peptide strands. Given that the absorbances of compounds in a mixture are cumulative, the observed absorbance being lower than expected indicates a hypochromic effect, suggesting an interaction between the components. The change in the absorbance and wavelength maximum in the UV spectra during the titration refers to the complex formation between proteins and PPIs. The Benesi–Hildebrand method was used to determine the stability constants based on Equation (7). The logarithms of the stability constants (*K* values) were compared, and the numerical data are available in Table 5. Some examples for fitting with corresponding r^2^ values can be found in Appendix A.

When analyzing the data obtained from UV titration, it is important to note that using a two-tailed *t*-test with a significance level of 0.05 and assuming equal variances did not reveal any significant difference in stability constants. The authors accepted the null hypothesis (H_0_), indicating no significant difference between the stability values, or that any observed difference could be due to chance. This suggests a limitation of the method, as it may not always be suitable for detecting small differences. Nevertheless, certain clear trends could still be observed. The molecules exhibited strong binding affinity toward both proteins. In all instances, AGP demonstrated a higher stability constant compared to HSA. The stability difference between the enantiomers was small, which is consistent with the bound percentage values determined by HPLC. However, even this small stability difference was sufficient for chiral separation in many cases. It should also be mentioned that during chromatographic analysis, the protein is bound to silica gel, whereas in spectroscopic examinations, it is in a free state.

### 2.3. HPLC and Fluorescence Displacement Study

For identification of the binding site of HSA, a displacement chromatography and fluorescence displacement study were performed. In the case of chromatography, to the mobile phase, increasing concentrations of the competitors, warfarin and ibuprofen, that are known to bind to specific binding sites on HSA, Sudlow site I and Sudlow site II, respectively, were added. In the case of the fluorescence displacement study, the fluorescence quenching data of the PPI-HSA system in the presence of warfarin and ibuprofen were investigated. The retenention factors obtained of each analyte in the absence and presence of the different concentrations of the competitors are depicted in Appendix A. A decrease in the *k* values with the increase in the concentration of the competitors was observed, independent of the analyte or the competitors used. It can also be seen that when ibuprofen was used as a competitor, the enantiomer recognition ability was lost, whereas it remained when warfarin was used (Appendix A). In previous literature studies using fluorescence spectroscopy, it is relatively univocal that PPIs bind to the Sudlow site I in subdomain II, which our fluorescence measurements also confirmed. More remarkable changes were observed in the presence of warfarin (Appendix A). Based on our data, it was also clear that Sudlow site I took part in the binding, which indicated a significant decrease in retention caused by warfarin. However, it is also clear that binding to Sudlow site II was responsible for the enantioseparation for lansoprazole. Based on these findings, we assumed that the main binding site was Sudlow site I for these compounds; however, the enantiomers could bind in different sites to a lesser extent, which could cause the enantioseparation for lansoprazole. A similar observation was made by Xu et al. for the analysis of omeprazole enantioseparation using HSA as a chiral selector by capillary electrophoresis [13].

### 2.4. Docking Study

Molecular docking was employed to calculate the binding strength of the studied PPI enantiomers (lansoprazole, rabeprazole, and omeprazole) toward AGP and HSA and analyze the molecular interactions at the atomic level. The calculated binding strength of the studied species toward AGP was nicely aligned with the HPLC data and able to explain the difference in enantiorecognition. It was found that in the case of rabeprazole and omeprazole, the *S* enantiomer could establish stronger interactions with the host (Figure 3). The docking scores were −7.4 kcal/mol and −6.7 kcal/mol for *S*-rabeprazole and *S*-omeprazole, respectively. These binding strengths were −0.3 and −0.8 kcal/mol stronger than in the case of the corresponding *R*-rabeprazole and *R*-omeprazole, respectively. For lansoprazole, the binding of the *R* enantiomer was favored by −0.4 kcal/mol over the *S* when the AGP was considered as the host.

If the interactions between the binding site of AGP and the ligands are considered, it can be seen that hydrogen bonds, π-π, and CH-π interactions were the most prevalent (Figure 3; Table 6).

The binding site of AGP is composed of four types of amino acid residues: tyrosine (TYR), arginine (ARG), phenylalanine (PHE), and glutamic acid (GLU). Three of these, ARG, GLU, and TYR, can establish hydrogen bonding interactions with the ligands, while PHE is involved in π-π and CH-π contacts. Four main interactions (two hydrogen bonds and two π interactions) were established in the case of the AGP/*R*-lansoprazole system, while only two hydrogen bonds were identified in the case of the AGP/*S*-lansoprazole pair, which is also in agreement with the abovementioned energetic preference toward the *R* enantiomer (Figure 3). The AGP/rabeprazole systems led to the formation of five main interactions for both the *R* and *S* enantiomers and, thus, the preference toward the latter cannot be explained just by considering the number of preferred contacts (Table 6; Figure 3). Furthermore, seemingly, *R*-rabeprazole established a stronger interaction with the binding site, as two hydrogen bonds were also formed. However, in the case of the *S* enantiomer, the formation of new π contacts with AGP were evident, and this, along with its more compact structure at the binding site, led to the preference toward this structure over the *R* antipode. In the case of omeprazole, the quantity of interactions formed in case of the *S* and *R* enantiomers agreed with the calculated docking score as well. As it can be seen, more interactions (five π and one hydrogen bonding) were established in the case of *S*-omeprazole compared to its *R* counterpart (Table 6; Figure 3).

In the case of HSA, five potential binding sites were identified using SiteMap (Appendix A) and used as target areas during the molecular docking calculations. Two of these, HSA-site1 and HSA-site2, are overlapping regions and, according to the literature, they are situated at the subdomain IIA of HSA, which is the region predicted to be the binding site [39,40,41]. This was also confirmed by fluorescence measurements in the current study (see above). Furthermore, when these two potential sites were selected as target regions for docking calculations, the resulting binding modes overlapped as well. Thus, HSA-site2 was selected to discuss interactions between binding of PPIs to human serum albumin. The docking scores for the five docking sites are summarized in Appendix A. The binding region included various amino acid residues, while the main interactions between HSA and the PPIs were established by tryptophan (TRP), leucine (LEU), lysine (LYS), alanine (ALA), arginine (ARG), and phenylalanine (PHE; Figure 4; Table 7).

Focusing on lansoprazole, it was observed that AGP had a preference toward the *R* enantiomer, while HSA preferably bound *S*-lansoprazole more strongly. This is in agreement with the earlier HPLC findings. By analyzing the structural features of the predicted binding modes of lansoprazole enantiomers by HSA, it can be seen that the favored contacts were the same (e.g., HC---π), but in the case of *R*-lansoprazole, an unfavored (CHxxxFC) interaction can also occur (Table 7), which could be the reasons behind the preference toward *S*-lansoprazole in the case of HSA (Figure 4; Table 7). Based on the docking score, it can be observed that the enantiomers of rabeprazole and omeprazole exhibited different binding strengths and modes when interacting with HSA. However, the disparity observed was not enough to achieve enantioseparation through the employed methodology.

All in all, most of the molecular docking results supported the experimental findings, especially considering the case of lansoprazole, and provided a detailed description of interactions between the hosts (AGP and HSA) and the studied proton pump inhibitor enantiomers.

## 3. Materials and Methods

### 3.1. Materials

Omeprazole, lansoprazole, rabeprazole sodium, warfarin, ibuprofen, and esomeprazole magnesium hydrate (*S*-omeprazole), together with the human plasma proteins, AGP (>99% agarose gel electrophoresis Sigma-Aldrich, St. Louis, Mo., USA,) and HSA (lyophilized powder, >96% agarose gel electrophoresis, Sigma-Aldrich, St. Louis, Mo., USA) were the product of Sigma-Aldrich, Hungary (Budapest, Hungary). Gradient-grade MeOH, EtOH, and IPA were purchased from Merck (Darmstadt, Germany). *R*-lansoprazole and *R*-rabeprazole were obtained from Beijing Mesochem Technology (Beijing, China). Analytical-grade dimethyl sulfoxide (DMSO), phosphate salts, phosphoric acid, ammonium acetate, ammonium bicarbonate, sodium citrate, and sodium hydroxide were procured from Sigma-Aldrich, Hungary (Budapest, Hungary). These chemicals were utilized for the formulation of the buffer solution. All reagents were employed without additional purification. Deionized water was generated using a Millipore Direct Q3 Millipore system (Millipore Corporation, Bedford, MA, USA). For the semi-preparative collection of *S*-lansoprazole, *S*-rabeprazole, and *R*-omeprazole, the following HPLC systems were used:

*R*-omeprazole: Chiralpak AD column (10 µm, 250 × 10 mm; Daicel Technologies, Tokyo, Japan) with neat MeOH, column temperature: 40 °C, and flow rate: 1.5 mL/min.

*S*-lansoprazole and *S*-rabeprazole: Lux Cellulose-4 column (5 µm, 150 × 4.6 mm; Phenomenex Torrance, CA, USA) with neat methanol, column temperature: 10 °C, and flow rate: 1.2 mL/min.

The purity of each enantiomer was assessed by reinjecting the samples into Lux Cellulose-4 (5 µm, 150 × 4.6 mm) and Lux Amylose-1 (5 µm, 150 × 4.6 mm) analytical columns using neat MeOH as an eluent, with a 1 mL/min flow rate and 25 °C column temperature. The enantiomeric purity exceeded 99.9% in each instance.

### 3.2. HPLC Measurements

Chromatographic measurements were carried out on two separate HPLC systems. The AGP binding study was performed on a Waters 2695e chromatographic system paired with a Waters 2996 PDA detector, controlled by Empower 3 FR5 (Waters Corp., Milford, MA, USA), while the HSA binding study was performed on an Agilent 1100 HPLC system, consisting of an inline degasser (G1322A), a quaternary pump (G1311A), an automatic injector (G1329A) paired with sample thermostat (G1330A), a column thermostat (G1316A), and a diode array detector (G1315A) with Agilent Chemstation B04.03-SP2 software (Agilent, Bronnwald, Germany). For the binding studies, two protein-based chiral stationary phases were utilized. The chromatographic binding study for AGP was conducted on a ChromTech Chiral-AGP column (150 mm × 4.0 mm; ChromTech Ltd., Congleton, UK), while the binding study for HSA was performed on a Daicel Chiralpak HSA column (50 mm × 4.0 mm; Daicel Corp., Osaka, Japan). The detection wavelength was 210 nm for all measurements. The stock solutions of PPIs were prepared at a concentration of 1 mg/mL in MeOH. Subsequent dilutions were prepared using water. The injected samples were mixed and diluted from the stock solutions. The final concentration for the eutomer was 0.2 mg/mL, and for the distomer, it was 0.1 mg/mL. A volume of 2 μL was injected. To mitigate the instability of PPIs in aqueous solutions, the samples were kept at a temperature of 4 °C. Cooling the samples to 4 °C prevented significant decomposition for a period of 24 h. After 24 h, new samples were prepared for injection. Chromatographic parameters were calculated based on the following equations. The retention factor (*k*) was calculated as:(1)$$k=\frac{\left(t_{r}-t_{0}\right)}{t_{0}}$$
where *t*_r_ is the retention time of the investigated analyte, and *t*_0_ is the dead time, determined as the first peak appearance in the chromatogram. Resolution was calculated based on the following equation:(2)$$R_{s}=2\frac{\left(t_{2}-t_{1}\right)}{\left(w_{1}+w_{2}\right)}$$
where t_2_ and t_1_ are the retention times of the second and first eluted enantiomers, respectively, while *w* is the peak width. The separation factor (*α*) was calculated as follows:(3)$$\alpha =\frac{k_{2}}{k_{1}}$$
where *k*_1_ and *k*_2_ are the retention factors of the first- and second-eluting enantiomers, respectively. The *k* values were also used to calculate the protein-bound percentage (*b%*) using Equation (4):(4)$$b\%=\frac{k}{1+k}\times 100$$

Classical van ’t Hoff analysis was applied to determine the thermodynamic parameters [59,60,61]. Van ’t Hoff plots were constructed by plotting the natural logarithm of the retention factor as a function of the inverse of the absolute temperature (thermodynamic temperature) over a temperature range of 10–30 °C:(5)$$\ln k=\frac{\Delta H{}^{\circ}}{RT}+\frac{\Delta S{}^{\circ}}{R}+\ln\Phi$$

In the equation, R represents the universal gas constant, T represents the absolute temperature in Kelvin, and *k* represents the retention factor of the enantiomers. Δ*H*° stands for the standard enthalpy, while Δ*S*° represents the standard entropy change of transfer of the solute from the mobile phase to the stationary phase. *Φ* is the phase ratio of the column. If Δ*H*° remains constant within the selected temperature range, a linear relationship can be observed between ln*k* and 1/T. The slope of this relationship is −Δ*H*°/R, and the intercept is Δ*S*°/R + ln Φ. Since the value of the phase ratio is rarely known, Δ*S*°* values (Δ*S*°* = Δ*S*° + Rln *Φ*) are often used to account for the uncertainty in *Φ*. Similarly, the difference in the change of standard enthalpy, Δ(Δ*H*°), and standard entropy, Δ(Δ*S*°), for the two enantiomers was calculated as follows:(6)$$\ln\alpha =-\frac{\Delta \left(\Delta H{}^{\circ}\right)}{RT}+\frac{\Delta \left(\Delta S{}^{\circ}\right)}{R}$$

### 3.3. Fluorescence Studies

Fluorescence spectra were measured using a Hitachi F-4500 fluorescence spectrophotometer (Hitachi, Tokyo, Japan) equipped with a xenon lamp source and 1.0 × 1.0 cm (3 mL) quartz cuvettes. The excitation slit was set to 5 nm and the emission slit to 10 nm. Excitation was applied at a wavelength of 295 nm, and emission spectra were recorded from 300 to 450 nm, with a maximum at 338 nm. Blank signals were subtracted to correct background noise. The blanks comprised a phosphate buffer with 1–7% DMSO at pH 7. Each experimental data point is the average of three independent experiments. The measurements were carried out at 24 °C. During the fluorescence titration, AGP and HSA were dissolved in a sodium phosphate buffer solution (pH 7.0; 0.1 M) to a concentration of 4 μM. The PPIs were dissolved and diluted in DMSO. PPI stock solutions were then further diluted with 0.1 M sodium phosphate solution (pH 7.0) to observe any fluorescence signals exhibited by the PPIs and in the presence of protein solution (4 μM). Proteins were titrated with the PPIs, resulting in final compound concentrations ranging from 0 to 140.74 μM. For correction purposes, the inner filter effect was assessed by recording the absorption spectra of each PPI. The absorption spectra were recorded on a Jasco V-750 spectrophotometer (Jasco, Tokyo, Japan) at 24 °C using 1.0 cm path-length quartz cuvettes. In the case of the fluorescence displacement study, HSA:warfarin and HSA:ibuprofen 1:1 solutions were titrated with the PPIs, resulting in final compound concentrations ranging from 0 to 19.58 μM (0, 1.99, 3.98, 5.96, 7.93, 9.89, and 19.58). All experiments were performed in triplicate.

### 3.4. UV Titration

The absorption spectra were recorded on a Jasco V-750 spectrophotometer (Jasco, Tokyo, Japan) at 24 °C using 1.0 cm path-length quartz cuvettes. Then, 1 μM HSA and AGP were dissolved in a sodium phosphate buffer solution (pH 7.4; 0.1 M) and their concentration was kept constant. The PPIs were dissolved in DMSO, and further diluted with the AGP or HSA solution. Next, 1000 μL of 1 μM HSA and AGP solutions was titrated with the PPI stock solution in 1 μL steps. The concentration of the PPIs was set to 4.00, 7.98, 11.96, 15.94, 19.90, 23.86, 27.81, and 31.75 μM. The binding constants were determined using the Benesi–Hildebrand equation:(7)$$\frac{A_{0}}{A-A_{0}}=\frac{\varepsilon_{PPI}}{\varepsilon_{APPI}-\varepsilon_{PPI}}+\frac{\varepsilon_{PPI}}{\varepsilon_{APPI}-\varepsilon_{PPI}}\times \frac{1}{K\times [C]}$$
where A_0_ is the absorbance of the HSA or AGP at 278 nm, A is the absorbance of the formed complex also at 278 nm, ε_PPI_ is the extinction coefficient of the actual PPI, ε_APPI_ is the extinction coefficient of the albumin or AGP and PPI complex, K is the binding coefficient, and [C] is the concentration of the investigated PPI. The plot of A_0_/A − A_0_ versus C^−1^ has a slope of K^−1^. All experiments were performed in triplicate.

### 3.5. In Silico Docking Studies

The Schrödinger suite (Release 2020-4) was employed to carry out the simulations. AGP and HSA structures (Protein Data Bank IDs: 3APW and 1H9Z, respectively) were used as host molecules during the docking calculations. In both cases, the Protein Preparation Wizard was employed [62], and the proteins were prepared by deleting non-protein species and using default settings. To predict the protonation state of the residues of the AGP and HSA models at physiological pH, PROPKA was employed [63], while to minimize the system, OPLS3e force field parameters were used [64]. In the case of HSA, SiteMap [65,66] was applied to identify potential binding sites, similar to our previous article [9]. For AGP, the location of the ligand from the PDB structure guided the docking. The LigPrep module with default settings was used to prepare the studied proton pump inhibitor structures, and all possible protonation states were considered at pH 7.0 ± 2.0. For each case, one enantiomer pair was selected and used as ligands in the docking. The Receptor Grid Generation module was employed to determine the grid for docking. The molecular docking calculations were carried out by Glide [67,68,69] using a flexible process and extra precision (XP) mode, and the complexes with the best docking scores were analyzed.

## 4. Conclusions

Proton pump inhibitors are frequently prescribed medications, and omeprazole is the eighth-most prescribed medication in the United States. Despite this, the enantioselective protein binding of these drugs has not previously been investigated. In this work, the enantioselective binding of lansoprazole, rabeprazole, and omeprazole to the two main transport proteins, AGP and HSA, was fully characterized using multimodal techniques, such as fluorescence, UV-Vis spectroscopy, HPLC, and molecular modeling. HPLC using protein-based columns is a great technique to detect enantioselective binding. The other modern stationary phases, such as the polysaccharide-based chiral column, can be used much easier for the detection of chiral impurity; however, the main advantage of the protein-based column is that it can be used for determination of the bound percentage. For stability constants’ determination, the spectroscopic method could be the best; however, it has some limitations, especially if the investigated compounds have their own fluorescence at the same wavelength as the protein. Molecular docking techniques can be used for characterization of the complex at the atomic level. For the exact characterization, more different techniques prove a more certain result; however, it should be noted that each technique has its own limitations and, sometimes, the comparison of the methods needs attention. Our work contributes to the understanding of the differences in pharmacokinetics between various enantiomers of PPIs. Based on the work, it was evident that enantiomers of PPIs bound to the transport proteins differently, which could be the relevance of the chiral switch regarding these compounds. Moreover, a stronger interaction was formed with AGP than albumin, which confirmed that the induced AGP levels in various pathological conditions can be an influencing factor on the pharmacokinetics of PPIs.

## Figures and Tables

**Figure 1 ijms-25-10575-f001:**
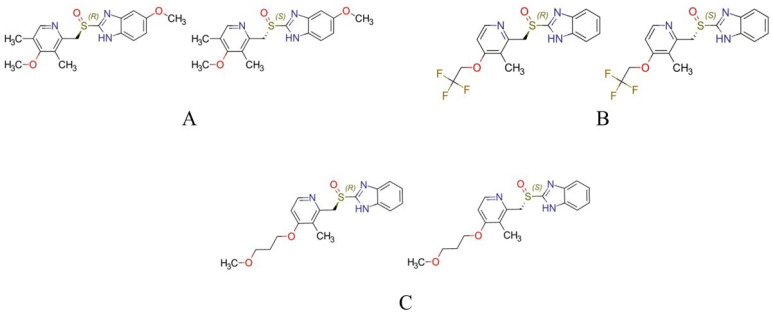
The structures of the investigated PPIs. (**A**) *R*-omeprazole (left) and *S*-omeprazole (right), (**B**) *R*-lansoprazole (left) and *S*-lansoprazole (right), and (**C**) *R*-rabeprazole (left) and *S*-rabeprazole (right).

**Figure 2 ijms-25-10575-f002:**
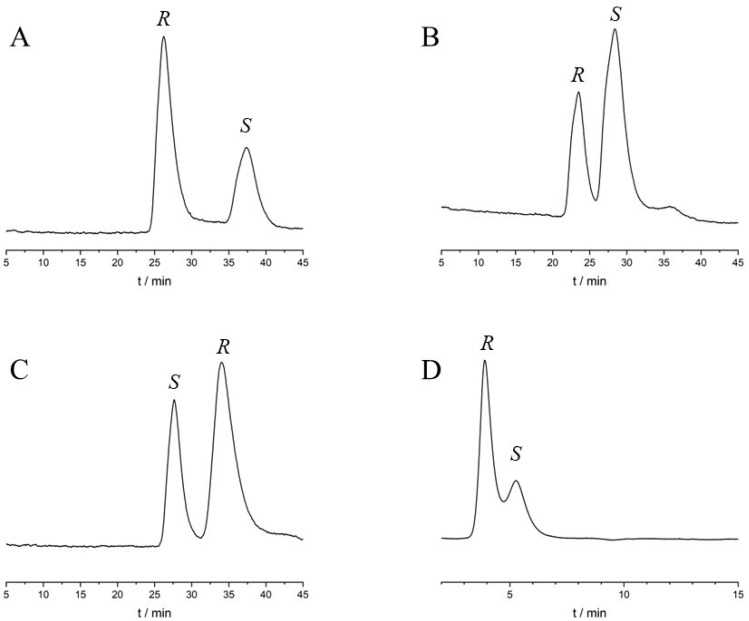
Chromatograms obtained during the HPLC study. (**A**) Rabeprazole, (**B**) omeprazole, and (**C**) lansoprazole on the ChromTech AGP column at 25 °C, mobile phase of 2.5% IPA and 97.5% 10 mM pH 7 sodium phosphate buffer, with a flow rate of 0.7 mL/min. (**D**) Chromatogram of lansoprazole on Chiralpak HSA at 25 °C, mobile phase of 2.5% IPA and 97.5% 10 mM pH 7 sodium phosphate buffer, with a flow rate of 0.7 mL/min. The detection wavelength was set at 210 nm.

**Figure 3 ijms-25-10575-f003:**
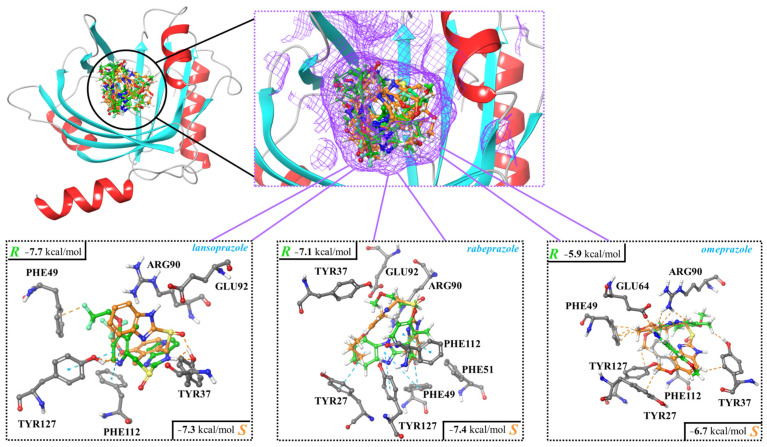
Interactions between human α1-acid glycoprotein (AGP) and enantiomers of lansoprazole, rabeprazole, and omeprazole. The complexes were determined using molecular docking, and the corresponding docking scores are also shown.

**Figure 4 ijms-25-10575-f004:**
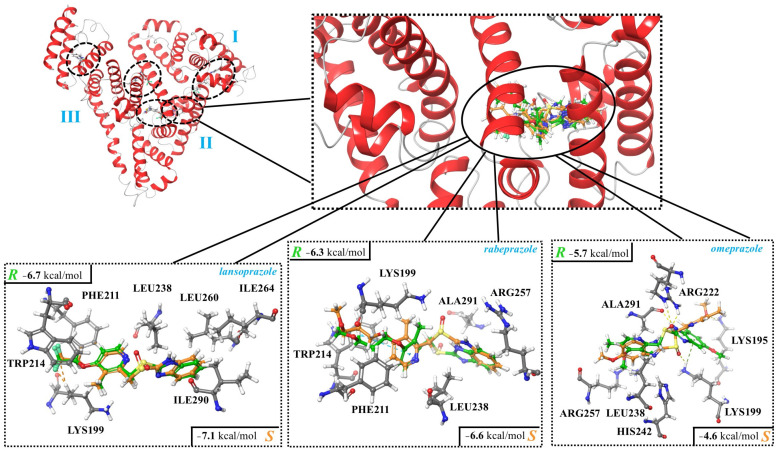
Interactions between human serum albumin (HSA; subdomain IIA) and the enantiomers of lansoprazole, rabeprazole, and omeprazole. The complexes were determined using molecular docking, and the corresponding docking scores are also shown.

**Table 1 ijms-25-10575-t001:** Previous works for characterizing PPI–protein complexes.

Investigated Compounds	Protein	Enantioselectivity	Method(s)	Main Result(s)	Ref.
Omeprazole, Pantoprazole, Ilaprazole	BSA	No	Fluorescence, UV-Vis, and circular dichroism (CD)	Omeprazole (log*K* = 4.58) Pantoprazole (log*K* = 5.04) Ilaprazole (log*K* = 5.63) Binding to subdomain IIA	[39]
Omeprazole	BSA	No	Fluorescence	*K* = 0.068 (µM^−1^)	[40]
Omeprazole and S-omeprazole	HSA	Partially *	Fluorescence, CD, voltametry, and in silico	Omeprazole (log*K* = 4.61) *S*-omeprazole (log*K* = 4.70) Binding to subdomain IIA	[41]
*R*-lansoprazole	HSA	Partially **	Fluorescence, UV, and molecular docking	log*K* = 3.44 Binding to subdomain IIA	[42]
*S*-omeprazole *R*-omeprazole	HSA	Yes	Affinity capillary electrophoresis and fluorescence	*R*-omeprazole (log*K* = 3.50) *S*-omeprazole (log*K* = 3.73) Enantioselective binding *S*-omeprazole bound to subdomain IIA, *R*-omeprazole bound to subdomain IIIA	[43]
Omeprazole and its metabolite	AGP	Yes	HPLC using AGP column	Stereoselective binding	[44]

* Esomeprazole was investigated but a stereoselective explanation was not published. ** Just one enantiomer (*R*-lansoprazole) was investigated.

**Table 2 ijms-25-10575-t002:** Retention factors, resolution values, and enantiomer elution order (EEO) in different percentages for different organic modifiers on the AGP and HSA columns at 25 °C. The applied buffer was a 10 mM pH 7 phosphate buffer.

Column	Modifier	%	Omeprazole				Rabeprazole				Lansoprazole			
			*k* _1_	*k* _2_	*R* _s_	EEO	*k* _1_	*k* _2_	*R* _s_	EEO	*k* _1_	k_2_	R_s_	EEO
AGP	IPA	2.5	13.24	16.18	1.2	*R < S*	14.89	21.66	2.5	*R < S*	15.75	19.61	1.5	*S < R*
		5	5.46	5.46	-	-	6.68	7.89	1.0	*R < S*	6.88	9.12	1.7	*S < R*
		7.5	2.88	2.88	-	-	3.901	3.90	-	-	4.00	5.28	1.5	*S < R*
		10	1.87	1.87	-	-	2.47	2.47	-	-	2.67	3.47	1.3	*S < R*
		12.5	1.33	1.33	-	-	1.72	1.72	-	-	1.95	2.47	1.1	*S < R*
		15	1.01	1.01	-	-	1.27	1.27	-	-	1.48	1.83	0.8	*S < R*
	EtOH	5	12.69	15.14	1.0	*R < S*	13.04	20.84	2.9	*R < S*	14.52	18.43	1.3	*S < R*
		7.5	6.57	7.20	0.2	*R < S*	7.21	10.50	2.0	*R < S*	8.24	10.64	1.3	*S < R*
		10	4.02	4.02	-	-	4.39	5.89	1.4	*R < S*	5.13	6.62	1.2	*S < R*
		15	1.81	1.81	-	-	2.11	2.48	0.6	*R < S*	2.54	3.21	1.0	*S < R*
	MeOH	5	25.74	37.33	1.8	*R < S*	20.89	42.82	3.7	*R < S*	22.63	28.55	1.2	*S < R*
		7.5	15.56	21.41	1.5	*R < S*	12.90	26.14	2.6	*R < S*	14.63	18.79	1.1	*S < R*
		10	9.95	13.05	1.1	*R < S*	8.63	16.71	3.1	*R < S*	9.96	12.56	1.1	*S < R*
		15	6.70	8.43	1.3	*R < S*	5.97	11.00	3.4	*R < S*	7.02	8.75	0.9	*S < R*
HSA	IPA	1	9.53	9.53	-	-	7.21	7.21	-	-	11.17	18.27	1.8	*R < S*
		2.5	8.66	8.66	-	-	5.45	5.45	-	-	9.26	12.87	1.0	*R < S*
		5	4.34	4.34	-	-	3.32	3.32	-	-	5.79	6.63	0.2	*R < S*
		7.5	2.82	2.82	-	-	2.37	2.37	-	-	4.37	4.37	-	-
		10	2.03	2.03	-	-	1.79	1.79	-	-	3.45	3.45	-	-
		12.5	1.50	1.50	-	-	1.37	1.37	-	-	2.71	2.71	-	-
	EtOH	1	9.89	9.89	-	-	6.16	6.16	-	-	10.13	14.32	1.0	*R < S*
		2.5	5.53	5.53	-	-	3.97	3.97	-	-	6.92	8.39	0.3	*R < S*
		5	3.68	3.68	-	-	2.87	2.87	-	-	5.26	5.26	-	-
		7.5	2.61	2.61	-	-	2.16	2.16	-	-	4.11	4.11	-	-
		10	1.97	1.97	-	-	1.66	1.66	-	-	3.24	3.24	-	-
	MeOH	1	7.79	7.79	-	-	6.61	6.61	-	-	10.66	15.61	1.0	*R < S*
		2.5	6.37	6.37	-	-	5.08	5.08	-	-	7.92	10.16	0.8	*R < S*
		5	4.95	4.95	-	-	3.68	3.68	-	-	6.05	7.29	0.2	*R < S*
		7.5	3.82	3.82	-	-	2.95	2.95	-	-	4.92	4.92	-	-
		10	2.97	2.97	-	-	2.34	2.34	-	-	4.03	4.03	-	-

**Table 3 ijms-25-10575-t003:** Calculated thermodynamic parameters, van ’t Hoff equation, correlation coefficients, Δ(ΔH°), Δ(ΔS°), Δ(ΔG°), T_iso_, and Q values.

Column	Compound	Equation	r^2^	Δ(ΔH°) (kJ/mol)	Δ(ΔS°) (J/molK)	Δ(ΔG°) (kJ/mol)	T_iso_ (°C)	Q
AGP	*S*-lansoprazole	3858.7x − 10.9	0.999	−32.1	−90.9	−5.0		
	*R*-lansoprazole	2681.1x − 6.3	0.999	−22.2	−52.7	−6.6		
		−1177.6x + 4.6	0.995	9.7	11.4	−1.6	−16.3	0.9
HSA	*R*-lansoprazole	1011.1x − 1.0	0.993	−8.4	−8.7	−5.8		
	*S*-lansoprazole	1357.9x − 1.9	0.996	−11.2	−15.9	−6.6		
		346.9x − 0.9	0.982	−2.8	−7.2	−0.7	128.9	1.4

**Table 4 ijms-25-10575-t004:** Bound percentage values (*b*%) determined for the extrapolation to zero percentages of different organic modifiers.

Column	Modifier	Omeprazole (%)		Rabeprazole (%)		Lansoprazole (%)	
		*S*	*R*	*S*	*R*	*S*	*R*
AGP	IPA	99.38	98.97	99.39	98.60	98.62	98.93
	EtOH	97.39	96.74	98.17	96.69	96.92	97.60
	MeOH	94.93	94.25	96.39	95.07	95.04	96.16
Average		97.23 ± 2.23	96.65 ± 2.36	97.98 ± 1.51	96.78 ± 1.77	96.86 ± 1.79	97.56 ± 1.39
HSA	IPA	91.15	91.15	88.24	88.24	93.67	91.77
	EtOH	92.97	92.97	88.85	88.85	94.85	92.69
	MeOH	91.03	91.03	89.42	89.42	95.42	92.89
Average		91.76 ± 1.08	91.76 ± 1.08	88.84 ± 0.59	88.84 ± 0.59	94.64 ± 0.89	92.45 ± 0.59

**Table 5 ijms-25-10575-t005:** The logarithms of the stability constants (log*K*) determined by UV titration (n = 3).

PPI	log*K*_AGP_ ± Stdev	log*K*_HSA_ ± Stdev.
*R*-omeprazole	4.69 ± 0.09	4.47 ± 0.30
*S*-omeprazole	4.47 ± 0.11	4.02 ± 0.45
*R*-lansoprazole	4.56 ± 0.28	4.22 ± 0.23
*S*-lansoprazole	4.83 ± 0.24	4.66 ± 0.14
*R*-rabeprazole	4.59 ± 0.03	4.33 ± 0.12
*S*-rabeprazole	4.76 ± 0.16	4.42 ± 0.15

**Table 6 ijms-25-10575-t006:** Interactions between the binding site of AGP and the PPI enantiomers, lansoprazole, rabeprazole, and omeprazole, predicted via molecular docking.

AGP/Lansoprazole		AGP/Rabeprazole		AGP/Omeprazole	
*R*	*S*	*R*	*S*	*R*	*S*
-TYR37 HO---H(N)-	-TYR37 HO---O=S-	-TYR37 OH---(O)-	n/a	-ARG90 (NH)---N-	-ARG90 (NH)---N-
-TYR127 OH---N-	n/a	-TYR127 OH---N-	n/a	-GLU64 (COO)---HN-	n/a
n/a	-GLU92 (COO)---HN	-TYR27 π---π	-TYR27 π---HC	n/a	-TYR27 π---HC
-PHE49 π---FC-	n/a	n/a	-TYR127 π---HC	n/a	-TYR37 π---π
-PHE112 π---HC	n/a	-PHE49 π---π	-PHE49 π ---HC	n/a	-TYR127 π---HC
-	-	n/a	-PHE51 CH---π	n/a	-PHE49 π---HC
-	-	-PHE112 π---π	-PHE112 π---π	n/a	-PHE112 π---HC

**Table 7 ijms-25-10575-t007:** Interactions between human serum albumin (HSA; subdomain IIA) and the studied proton pump inhibitor enantiomers, lansoprazole, rabeprazole, and omeprazole, predicted via molecular docking (--- marking favored and xxx marking unfavored interactions).

HSA/Lansoprazole		HSA/Rabeprazole		HSA/Omeprazole	
*R*	*S*	*R*	*S*	*R*	*S*
-TRP214 π ---FC-	-TRP214 π---FC-	-TRP214 π---π	-TRP214 π---HC-	-LYS199 -NH---π	n/a
-LEU238 HC---π	-LEU238 HC---π	n/a	-PHE211 π --- π	-ALA291 (C=O)---HN	n/a
-LYS199 -CHxxxFC-	-	-LYS199 -CHxxxHC-	-LYS199 -CH xxx HC-	-ARG222 (NH)---(O=S)	n/a
-	-	-LYS199 -NH xxx HC-	n/a	-LEU238 CH---π	-LEU238 CH---π
-	-	-PHE211 -CHxxxHC-	-PHE211 -CHxxxHC-	n/a	-LYS195 -NH----O-
-	-	-	-	n/a	-LYS199 -NH---(O=S)
-	-	-	-	n/a	-ALA291 CH---π

## Data Availability

The original contributions presented in this study are included in the article/Appendix A. Further inquiries can be directed to the corresponding author.

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
