# Peer review of "Enantioselective Binding of Proton Pump Inhibitors to Alpha1-Acid Glycoprotein and Human Serum Albumin—A Chromatographic, Spectroscopic, and In Silico Study"

_ijms, 2024, doi:10.3390/ijms251910575_

Round 1

Reviewer 1 Report

Comments and Suggestions for Authors

Comments:

1.      The Abstract section should

a) clearly identify the research question, the main work scope, methodology and quantitative results. The abstract would greatly benefit from reorganization of thoughts.

b) avoid the use of subjective terms and modify with more quantitative results.

2.      The Introduction section provides

a) more information on the current research status of PPIs binding to proteins in previous studies, as well as the novelty and improvements of this study compared to existing knowledge.

b) Emphasize the advantages and potential application prospects of the combined use of multiple methods in revealing the binding characteristics of PPIs to proteins in this study.

c) Clearly state the important role of enantioselective binding research in drug development, especially in the pharmacokinetic and pharmacodynamic studies of chiral drugs.

d) provide a relevant reference, such as sentence ‘In healthy individuals, the basal plasma concentration of AGP typically is around 20 μM. However, in conditions characterized by stress, such as sepsis, this concentration can surge dramatically increasing up to fivefold.’

3.      The discussion section of the 2.2 Spectral study

a) lacks quenching spectra and fitting curve spectra for spectral experiments, and

b) lacks the result table does not summarize the degree of fitting (R2). This part is very important for the analysis of spectral experimental results and cannot be omitted.

c) Besides, the fluorescence emission spectrum of the maximum blank daphnetin under experimental conditions should be added to eliminate the influence of self-fluorescence on the protein emission spectrum.

d) More relevant research should be cited and discussed regarding the fluorescence section, such as: 10.26599/fshw.2022.9250033; 10.1016/j.ijbiomac.2018.02.137; 10.1021/acs.jpcb.2c06488.

4.      The paper is lacking a hypothesis. Please correct.

5.      The statistics methods are unclear. Which tests were used.

6.      What is the repeatability of the obtained test results?

Comments on the Quality of English Language

Grammar revision is required.

Author Response

  1. The Abstractsection should
  2. a) clearly identify the research question, the main work scope, methodology and quantitative results. The abstract would greatly benefit from reorganization of thoughts.
  3. b) avoid the use of subjective terms and modify with more quantitative results.

Thank you for your comment. Based on your remarks, the abstract was modified as follows:

The enantioselective binding of three proton pump inhibitors (PPIs) – omeprazole, rabeprazole, and lansoprazole – to two key plasma proteins, α1-acid glycoprotein (AGP) and human serum albumin (HSA) were characterized. The interactions between PPI enantiomers and proteins were investigated by using a multifaceted analytical approach, including high-performance liquid chromatography (HPLC), fluorescence- and UV spectroscopy as well as in silico molecular docking. HPLC analysis demonstrated that all three PPIs exhibited enantioseparation on an AGP-based chiral stationary phase, suggesting stereoselective binding to AGP, while only lansoprazole showed enantioselective binding on the HSA-based column. Quantitatively, the S-enantiomers of omeprazole and rabeprazole showed higher binding affinity to AGP, while the R-enantiomer of lansoprazole displayed greater affinity for AGP, with a reversal in elution order observed between the two protein-based columns. Protein binding percentages, calculated via HPLC, were greater than 88% for each enantiomer across both transport proteins, with all enantiomers displaying higher affinity for AGP compared to HSA. Thermodynamic analysis indicated on the HSA the more common, enthalpy-controlled, while in contrast, on the AGP entropy-controlled enantioseparation was observed. The study also identified limitations in using fluorescence titration due to the high native fluorescence of the compounds, whereas UV titration was effective for both proteins. The determined logK values are in the range of 4.47 – 4.83 for AGP and 4.02 – 4.66 for HSA. Molecular docking supported the experimental findings by revealing the atomic interactions driving the binding process, with predicted enantiomer elution orders aligning with experimental data. The comprehensive use of these analytical methods provides detailed insights into the enantioselective binding properties of PPIs, contributing to the understanding of their pharmacokinetic differences and aiding in the development of more effective therapeutic strategies.

  1. The Introductionsection provides
  2. a) more information on the current research status of PPIs binding to proteins in previous studies, as well as the novelty and improvements of this study compared to existing knowledge.

Table 1 contains the main information from previous studies. We extended the revised version of the manuscript to include the current research status of PPI binding to proteins. This information clearly shows that the enantioselective binding of PPIs is underrepresented in the literature. Therefore, it should be investigated more deeply.

Line 159-172 in the revised version of the manuscript was modified as follows:

Only two works deal with the enantioselective protein binding of PPIs. Karlsson and Hermansson developed a chiral HPLC method for enantioseparation of omeprazole and its metabolite on a chiral AGP-based column. Based on the successful enantioseparation, enantioselective binding can be implied, however, the study was not designed to map the binding of the studied enantiomers to the drug-binding transport protein [44]. In another work, Xu et. al developed a method for the separation of omeprazole enantiomers by affinity capillary electrophoresis using HSA as chiral selector, clearly indicating the HSA binding of the analyte is enantioselective, which was further proved by fluorescence measurements [43]. The authors also investigated the location of the binding site through displacement studies using warfarin and ibuprofen as markers for subdomain IIA (Sudlow site I) and subdomain IIIA (Sudlow site II). Based on the results, esomeprazole is more likely to bind to Sudlow site I in subdomain IIA. However, it appears that R-omeprazole may bind to a different region of the protein as well. A comparison of our results with those from previous literature can be found in Supplementary Table 1.

  1. b) Emphasize the advantages and potential application prospects of the combined use of multiple methods in revealing the binding characteristics of PPIs to proteins in this study.

Thank you for your remarks. First it should be noted that a complex analytical question – like an enantioselective binding – needs a multi-analytical approach using a wide variety of methods. By using HPLC, fluorescence, UV spectroscopy, and in silico molecular docking, the study provides a thorough understanding of the enantioselective binding characteristics of PPIs. Each method contributes unique insights — for example, HPLC confirms the stereoselective binding of PPIs to AGP and HSA, the spectroscopic method is excellent for correct determination of stability constants, while molecular docking reveals atomic-level interactions and the driving forces of these bindings. This multi-analytical approach allows for a more nuanced understanding of the binding processes than any single method could offer. The following part was added to the revised version of the manuscript (line 116-123):

Characterizing protein-ligand interactions can only be effectively accomplished using a multi-analytical approach, which takes advantages of each analytical method and grants that the results obtained are validated through different techniques. For the rapid characterization of enantioselective binding, methods such as HPLC or electrophoresis can be applied. Spectroscopic techniques are suitable for determining stability constants, while computational modeling is used to describe interactions at the atomic level.

  1. c) Clearly state the important role of enantioselective binding research in drug development, especially in the pharmacokinetic and pharmacodynamic studies of chiral drugs.

The different binding characteristics of enantiomers can lead to significant variations in a drug's absorption, distribution, metabolism, and excretion (ADME), as well as its therapeutic efficacy and safety profile. It is well-established that enantiomers can exhibit distinct therapeutic effects; a classic example is thalidomide, where R-thalidomide has sedative effects, while S-thalidomide is teratogenic. In the case of PPIs, enantioselective pharmacokinetics is particularly important, as these molecules bind to their target receptors following an achiral transformation. The enantioselective binding of PPIs to plasma proteins results in different pharmacokinetic profiles for each enantiomer. Understanding enantioselective binding can aid in selecting the most appropriate enantiomer with favorable pharmacokinetic properties, optimizing both efficacy and safety in drug development. The revised version of the manuscript was modified as follows (line 123-131):

The different binding characteristics of enantiomers can lead to significant variations in a drug's absorption, distribution, metabolism, and excretion (ADME), as well as its therapeutic efficacy and safety profile. It is well-established that enantiomers can exhibit distinct therapeutic effects; a classic example is thalidomide, where R-thalidomide has sedative effects, while S-thalidomide is teratogenic. In addition to enantioselective binding to the target molecule, enantioselective binding to transport proteins is also important because it provides insights into how different enantiomers are distributed, transported, and cleared in the body. This understanding enables the selection of the most effective and safe enantiomer, optimizing therapeutic outcomes and minimizing potential side effects.

  1. d) provide a relevant reference, such as sentence ‘In healthy individuals, the basal plasma concentration of AGP typically is around 20 μM. However, in conditions characterized by stress, such as sepsis, this concentration can surge dramatically increasing up to fivefold.’

Based on your request the following reference added to the questionable section:

 Ceciliani, F.; Lecchi, C. The Immune Functions of α Acid Glycoprotein. Curr Protein Pept Sc 2019, 20, 505-524, doi:10.2174/1389203720666190405101138.

  1. The discussion section of the 2.2 Spectral study
  2. a) lacks quenching spectra and fitting curve spectra for spectral experiments, and
  3. b) lacks the result table does not summarize the degree of fitting (R2). This part is very important for the analysis of spectral experimental results and cannot be omitted.
  4. c) Besides, the fluorescence emission spectrum of the maximum blank daphnetin under experimental conditions should be added to eliminate the influence of self-fluorescence on the protein emission spectrum.

Based on the reviewer remarks Supplementary Materials were created, incorporating the following: quenching spectra, blank spectra, the spectra of PPIs and proteins. Some examples for fitting curves with r2 values were also added. It should be noted that all experiments were performed in triplicate. Table 5 is supplemented with this information.  The Supplementary Materials contain the following:

Table of Contents

Supplementary Table 1. Comparison of recent study results with previous literature

Supplementary Figure 1. Chromatograms obtained using different buffer constituents on HAS column: A – 10 mM sodium phosphate, pH 7; B – 10 mM ammonium acetate, pH 7; C – 10 mM ammonium acetate, pH 7; D – 10 mM sodium citrate, pH 7. The column temperature is 25 °C, and the applied organic modifier is 1% 2-propanol

Supplementary Table 2. Retention times and resolution values using different buffer constituents on the AGP and HSA columns at 25 °C. The applied organic modifier is 2.5% 2-propanol for the AGP column and 1% 2-propanol for the HSA column

Supplementary Figure 2. Representative spectra obtained from fluorescence measurements: A – Unprocessed fluorescence spectra of R-lansoprazole and AGP; B – Unprocessed fluorescence spectra of S-lansoprazole and AGP; C – Emission spectra of the investigated compounds; D – Background fluorescence of the matrix (phosphate buffer).

Supplementary Table 3. Absorbance values used for the inner filter effect calculation for lansoprazole. The absorbance values for the specified solution compositions were measured with UV-Vis spectrophotometer as part of fluorescence quenching studies, with an excitation wavelength of 295 nm and an emission wavelength of 340 nm.

Supplementary Table 4. The observed and IFE corrected fluorescence intensity values for S-lansoprazole and R-lansoprazole, respectively.

Supplementary Figure 3. Stern-Volmer plots for S-lansoprazole and R-lansoprazole. Although Stern-Volmer plots typically exhibit an increasing trend, the scatter plots here show minimal linearity in the data points (r² = 0.3371; r² = 0.4126).

Supplementary Figure 4. Representative UV spectra: A – R-lansoprazole and AGP (1 μM); B – S-lansoprazole and AGP (1 μM); C – R-rabeprazole and AGP (1 μM); D – S-rabeprazole and AGP (1 μM)

Supplementary Figure 5. Examples of fitting the Benesi-Hildebrandt equation. The r² value is higher than 0.995 in all cases: A – S-rabeprazole (r² = 0.9994); B – R-omeprazole (r² = 0.9979); C – S-omeprazole (r² = 0.9979).

Supplementary Table 5. Changes in retention factor values in the presence of warfarin and ibuprofen at different concentrations on HSA column. (Chromatographic parameter: Column temperature: 25 °C, mobile phase: 2-propanol : 10 mM pH 7 sodium phosphate buffer 1:99, flow rate: 0.7 mL/min, detection wavelength: 210 nm).

Supplementary Figure 6. The fluorescence spectra of HSA-warfarin complex titrated with S-omeprazole

Supplementary Figure 7. The potential binding sites in HSA identified by Sitemap

Supplementary Table 6.  Docking scores of PPIs at potential binding sites of HSA

  1. d) More relevant research should be cited and discussed regarding the fluorescence section, such as: 10.26599/fshw.2022.9250033; 10.1016/j.ijbiomac.2018.02.137; 10.1021/acs.jpcb.2c06488.

The references were added to the revised version of the manuscript as reference 10,55,56.

  1. The paper is lacking a hypothesis. Please correct.

We hypothesize that the enantiomers of PPIs exhibit different binding affinities to AGP and HSA, leading to enantioseparation on protein-based columns. These differences in binding affinities can be characterized using a range of techniques, including those that provide atomic-level insights. The distinct binding characteristics are expected to account for the variations in the ADME parameters of each enantiomer.

  1. The statistics methods are unclear. Which tests were used.

The UV part modified as follows:

When analyzing the data obtained from UV titration, it is important to note that using a two-tailed t-test with a significance level of 0.05 and assuming equal variances did not reveal any significant difference in stability constants. The authors accepted null hypothesis (H0), indicating no significant difference between the values of K, or that any observed difference could be due to chance. () This suggests a limitation of the method, as it may not always be suitable for detecting small differences.

  1. What is the repeatability of the obtained test results?

All experiments were performed in triplicate.

Reviewer 2 Report

Comments and Suggestions for Authors

Recommendation: Publish after major revisions noted.

Comments:

This manuscript from GergÅ‘ Tóth et. al reported the stereoselective interactions between omeprazole, lansoprazole, and rabeprazole, and the primary human transport proteins, HSA and AGP. Multimodal techniques, including chromatographic, spectroscopic, and molecular modeling approaches, were employed to ensure reliable results. This work contributes to the understanding of the pharmacokinetic differences between different PPI enantiomers. However, there are some questions while I read the current manuscript. Thus I recommended a major revision is required before this work has been published on International Journal of Molecular Sciences.

(1) Have the authors investigated how changing salt solution type and concentration affects the protein binding?

(2) The binding site markers (Warfarin and Ibuprofen) are usually used to investigate the binding site between drugs and proteins. Some competitive combination studies are suggested to be added to illustrate the binding site between those proton pump inhibitors with AGP and HAS, respectively.

(3) The representative UV-vis spectra of PPIs with AGP or HAS are suggested to be added in the manuscript. It can help readers to learn about the changes in UV-visible absorption at characteristic peak in difference systems.

(4) What molecular docking method did the author use? Rigid docking, semi-flexible docking, or flexible docking?

(5) The authors are suggested to check the format of the references carefully. For example, the pages should be added in reference 1 and 6.

Comments on the Quality of English Language

 Minor editing of English language required.

Author Response

This manuscript from GergÅ‘ Tóth et. al reported the stereoselective interactions between omeprazole, lansoprazole, and rabeprazole, and the primary human transport proteins, HSA and AGP. Multimodal techniques, including chromatographic, spectroscopic, and molecular modeling approaches, were employed to ensure reliable results. This work contributes to the understanding of the pharmacokinetic differences between different PPI enantiomers. However, there are some questions while I read the current manuscript. Thus I recommended a major revision is required before this work has been published on International Journal of Molecular Sciences.

  • Have the authors investigated how changing salt solution type and concentration affects the protein binding?

Based on this request three other buffers were also investigated (ammonium-acetate, ammonium-bicarbonate and sodium-citrate).  Representative chromatograms and the main data were added to the Supplementary Materials of the revised version of the manuscript (Supplementary Figure 1, and Supplementary Table 2). Overall, the peak shape is the best in phosphate buffer, while using other types of salts, the peak shape deteriorates and becomes distorted. The retention time is similar in all buffers, the resolution values show slight variations. Due to the better peak shape and appropriate resolution, phosphate buffer was chosen for further study.

The revised version of the manuscript was supplemented as follows (line 234-242):

The effect of different buffer constituents was also evaluated. In addition to phosphate buffer, ammonium acetate, ammonium bicarbonate, and sodium citrate were tested and compared under the same conditions. The representative chromatograms and the principal chromatographic results are presented in Supplementary Figure 1 and Supplementary Table 2, respectively. In general, the peak shape was optimal with phosphate buffer, while the use of other salts resulted in peak distortion. Although the retention times were similar across all buffers, slight variations were observed in resolution values. Due to the superior peak shape and adequate resolution, a phosphate buffer was selected for further studies.

(2) The binding site markers (Warfarin and Ibuprofen) are usually used to investigate the binding site between drugs and proteins. Some competitive combination studies are suggested to be added to illustrate the binding site between those proton pump inhibitors with AGP and HAS, respectively.

Thank you for your comment. Based on your request, HPLC and fluorescence displacement studies were performed. The revised version of the manuscript was supplemented as follows (line 362-384):

2.3 HPLC and fluorescence displacement study

For identification of the binding site of HSA a displacement chromatography and fluorescence displacement study were performed. In the case of chromatography to the mobile phase increasing concentrations of the competitors, warfarin and ibuprofen, that are known to bind to specific binding sites on HSA, Sudlow site I and Sudlow site II, respectively was added. In the case of fluorescence displacement study, the fluorescence quenching data of PPI-HSA system in the presence of warfarin and ibuprofen were investigated.   The k values obtained of each analyte in the absence and presence of the different concentrations of the competitors are depicted in Supplementary Table 5. A decrease in the k values with the increase in the concentration of the competitors was observed, independent of the analyte or the competitors used. It can also be seen that when ibuprofen is used as a competitor, the enantiomer recognition ability is lost, whereas it remains when warfarin is used (Supplementary Figure 6). Previous literature works using fluorescence spectroscopy is relatively univocal, that PPIs bind to the Sudlow site I in subdomain II, which our fluorescence measurements also confirmed. More remarkable changes were observed in the presence of warfarin (Supplementary Figure 7). Based on our data it is also clear that Sudlow site I take a part in the binding, that indicates a significant decrease in retention caused by warfarin. However, it is also clear that binding to Sudlow site II is responsible for the enantioseparation for lansoprazole. Based on these we assume that the main binding site is Sudlow site I for these compounds, however the enantiomers could bind in different sites in less extent, that could cause the enantioseparation for lansoprazole. Similar observation was taken by Xu et. al for analysis of omeprazole enantioseparation using HSA as chiral selector by capillary electrophoresis [13].

(3) The representative UV-vis spectra of PPIs with AGP or HAS are suggested to be added in the manuscript. It can help readers to learn about the changes in UV-visible absorption at characteristic peak in difference systems.

Based on the reviewer’s comment fluorescence quenching and UV-Vis spectra were added to the Supplementary Materials. The Supplementary Materials include the following items:

Table of Contents

Supplementary Table 1. Comparison of recent study results with previous literature

Supplementary Figure 1. Chromatograms obtained using different buffer constituents on HAS column: A – 10 mM sodium phosphate, pH 7; B – 10 mM ammonium acetate, pH 7; C – 10 mM ammonium acetate, pH 7; D – 10 mM sodium citrate, pH 7. The column temperature is 25 °C, and the applied organic modifier is 1% 2-propanol

Supplementary Table 2. Retention times and resolution values using different buffer constituents on the AGP and HSA columns at 25 °C. The applied organic modifier is 2.5% 2-propanol for the AGP column and 1% 2-propanol for the HSA column

Supplementary Figure 2. Representative spectra obtained from fluorescence measurements: A – Unprocessed fluorescence spectra of R-lansoprazole and AGP; B – Unprocessed fluorescence spectra of S-lansoprazole and AGP; C – Emission spectra of the investigated compounds; D – Background fluorescence of the matrix (phosphate buffer).

Supplementary Table 3. Absorbance values used for the inner filter effect calculation for lansoprazole. The absorbance values for the specified solution compositions were measured with UV-Vis spectrophotometer as part of fluorescence quenching studies, with an excitation wavelength of 295 nm and an emission wavelength of 340 nm.

Supplementary Table 4. The observed and IFE corrected fluorescence intensity values for S-lansoprazole and R-lansoprazole, respectively.

Supplementary Figure 3. Stern-Volmer plots for S-lansoprazole and R-lansoprazole. Although Stern-Volmer plots typically exhibit an increasing trend, the scatter plots here show minimal linearity in the data points (r² = 0.3371; r² = 0.4126).

Supplementary Figure 4. Representative UV spectra: A – R-lansoprazole and AGP (1 μM); B – S-lansoprazole and AGP (1 μM); C – R-rabeprazole and AGP (1 μM); D – S-rabeprazole and AGP (1 μM)

Supplementary Figure 5. Examples of fitting the Benesi-Hildebrandt equation. The r² value is higher than 0.995 in all cases: A – S-rabeprazole (r² = 0.9994); B – R-omeprazole (r² = 0.9979); C – S-omeprazole (r² = 0.9979).

Supplementary Table 5. Changes in retention factor values in the presence of warfarin and ibuprofen at different concentrations on HSA column. (Chromatographic parameter: Column temperature: 25 °C, mobile phase: 2-propanol : 10 mM pH 7 sodium phosphate buffer 1:99, flow rate: 0.7 mL/min, detection wavelength: 210 nm).

Supplementary Figure 6. The fluorescence spectra of HSA-warfarin complex titrated with S-omeprazole

Supplementary Figure 7. The potential binding sites in HSA identified by Sitemap

Supplementary Table 6.  Docking scores of PPIs at potential binding sites of HSA

(4) What molecular docking method did the author use? Rigid docking, semi-flexible docking, or flexible docking?

 Thank you for your question. We have employed a flexible docking procedure during the calculations. The methods section has been modified to clarify this point:

" The molecular docking calculations were carried out by Glide [67-69] using a flexible docking procedure and extra precision (XP) mode (...)"

(5) The authors are suggested to check the format of the references carefully. For example, the pages should be added in reference 1 and 6.

Reference format was checked and modified according to the journal rules.

Reviewer 3 Report

Comments and Suggestions for Authors

The study is complex, quite well-written, and addresses a topic of interest for the journal. The theme aligns with the Aims and Scope, but there are a few aspects that need to be clarified or revised by the authors before the manuscript can be recommended for publication.

Observations and comments:

  • The data from the present study should be added to Table 1 to better highlight the degree of novelty and originality. These aspects are not well emphasized in the introduction, and including them in the comparative table with the literature data would be beneficial in this regard.

  • The practical importance of the study is not very well highlighted. The usefulness of all the methods used is not clear, especially since they did not yield similar results when employed for the same purpose. The authors are requested to better justify the choice of methods used and the role of each, as well as the importance of the study and its field of applicability.

  • In the Results and Discussion section, experimental details related to the working conditions used are presented, which should be moved to section 3 of the manuscript, where the respective methods and procedures are discussed.

Author Response

  • The data from the present study should be added to Table 1 to better highlight the degree of novelty and originality. These aspects are not well emphasized in the introduction, and including them in the comparative table with the literature data would be beneficial in this regard.

We greatly appreciate your insightful comments. In response to Reviewer 1's remarks, we have revised the introduction section. Here the novelty of our work was further highlighted, and some previous findings were summarized. We believe that placing the summary of our results in the introduction section would not be ideal; however, in line with your suggestions, we have prepared a new table that presents both the previous literature data and our novel findings. This new table has been added to the Supplementary Material, as Supplementary Table 1.

  • The practical importance of the study is not very well highlighted. The usefulness of all the methods used is not clear, especially since they did not yield similar results when employed for the same purpose. The authors are requested to better justify the choice of methods used and the role of each, as well as the importance of the study and its field of applicability.

We appreciate your comments. We have revised the manuscript to further clarify the value and applicability of our research. The primary importance of our study lies in its comprehensive approach for the characterization of the enantioselective binding of PPIs to AGP and HSA. Understanding these interactions is crucial for optimizing drug design, improving therapeutic efficacy, and minimizing adverse effects as well as helping to find the enantiomers with better pharmacokinetic properties. We employed a multianalytical approach (HPLC, fluorescence, UV spectroscopy, and molecular docking) to gain an in-depth understanding of these complex interactions. Each technique has a specific role in the characterization of these interactions: HPLC allows for rapid enantioseparation, providing initial insights into enantioselective binding. Fluorescence and UV spectroscopy offer means to quantify the binding affinities and stability constants, each overcoming specific limitations of the other. UV spectroscopy, for example, avoids interference from the native fluorescence of the PPIs. Molecular docking provides atomic-level details of the binding interactions, validating and explaining the experimental observations.

Although these methods did not yield identical results, this diversity reflects the complexity of enantioselective interactions in different environments, highlighting the need for a multifaceted approach. Our study demonstrates that employing multiple analytical methods provides a more comprehensive understanding of drug-protein interactions,

We have now emphasized these points in the revised manuscript to better communicate the study's practical significance.

  • In the Results and Discussion section, experimental details related to the working conditions used are presented, which should be moved to section 3 of the manuscript, where the respective methods and procedures are discussed.

Based on this request some experimental parameters moved to section 3.

Round 2

Reviewer 2 Report

Comments and Suggestions for Authors

The revised manuscript has been carefully improved, and it can be published on IJMS.

Comments on the Quality of English Language

Minor editing of English language required.